# ECG Localization Method Based on Volume Conductor Model and Kalman Filtering

**DOI:** 10.3390/s21134275

**Published:** 2021-06-22

**Authors:** Yuki Nakano, Essam A. Rashed, Tatsuhito Nakane, Ilkka Laakso, Akimasa Hirata

**Affiliations:** 1Department of Electrical and Mechanical Engineering, Nagoya Institute of Technology, Nagoya 466-8555, Japan; y.nakano.154@nitech.jp (Y.N.); essam.rashed@nitech.ac.jp (E.A.R.); t.nakane.901@nitech.jp (T.N.); 2Department of Mathematics, Faculty of Science, Suez Canal University, Ismailia 41522, Egypt; 3Department of Electrical Engineering and Automation, Aalto University, 02150 Espoo, Finland; ilkka.laakso@aalto.fi; 4Center of Biomedical Physics and Information Technology, Nagoya Institute of Technology, Nagoya 466-8555, Japan

**Keywords:** electrocardiography, cardiac source localization, finite difference methods, inverse problems

## Abstract

The 12-lead electrocardiogram was invented more than 100 years ago and is still used as an essential tool in the early detection of heart disease. By estimating the time-varying source of the electrical activity from the potential changes, several types of heart disease can be noninvasively identified. However, most previous studies are based on signal processing, and thus an approach that includes physics modeling would be helpful for source localization problems. This study proposes a localization method for cardiac sources by combining an electrical analysis with a volume conductor model of the human body as a forward problem and a sparse reconstruction method as an inverse problem. Our formulation estimates not only the current source location but also the current direction. For a 12-lead electrocardiogram system, a sensitivity analysis of the localization to cardiac volume, tilted angle, and model inhomogeneity was evaluated. Finally, the estimated source location is corrected by Kalman filter, considering the estimated electrocardiogram source as time-sequence data. For a high signal-to-noise ratio (greater than 20 dB), the dominant error sources were the model inhomogeneity, which is mainly attributable to the high conductivity of the blood in the heart. The average localization error of the electric dipole sources in the heart was 12.6 mm, which is comparable to that in previous studies, where a less detailed anatomical structure was considered. A time-series source localization with Kalman filtering indicated that source mislocalization could be compensated, suggesting the effectiveness of the source estimation using the current direction and location simultaneously. For the electrocardiogram R-wave, the mean distance error was reduced to less than 7.3 mm using the proposed method. Considering the physical properties of the human body with Kalman filtering enables highly accurate estimation of the cardiac electric signal source location and direction. This proposal is also applicable to electrode configuration, such as ECG sensing systems.

## 1. Introduction

An electrocardiogram (ECG) is an essential tool for the early detection of heart disease [1]. A 12-lead ECG displays the records of signals expanded to 12 patterns based on the potentials measured at standardized electrode positions: six electrodes are located on the patient’s chest, and one electrode is attached to each arm and left leg [1]. The ECG waveforms record a combination of electrical activity from various cardiac cells; a typical waveform consists of three phases: P-wave, QRS-complex, and T-wave. Much attention has been paid to the classification of diseases and localization of cardiac sources with ECG waveform (e.g., [2,3,4,5]). By estimating the time-varying source of the electrical activity from the potential changes, several types of heart disease can be noninvasively identified. However, most previous studies are based on signal processing, and thus additional approach based on physics would be helpful, especially for localization.

The localization of the cardiac source based on a 12-lead ECG is a well-known ill-posed problem [6]. This is because only nine observation points are considered, whereas the unknown or potential source locations/directions are substantial. A unique solution to the source localization problem cannot be exactly determined. Moreover, it is difficult to model the observed potential in a straightforward manner because of the inhomogeneity of the human body, which is composed of tissues with different electrical conductivity values. Therefore, the development of a cardiac source localization technique using a 12-lead ECG is an essential research topic for several clinical applications.

Previous studies have been conducted to solve the inverse problem of electrocardiography [7,8,9]. In these studies, a forward problem analysis was conducted using geometric conductor models representing a human torso or realistic torso model. However, misplacement of the electrode positions causes substantial errors in clinical ECG signals [10]. Moreover, the position of the limb electrodes directly influences all leads in terms of the shape and amplitude of the ECG waveform [11]. Thus, to reproduce a realistic 12-lead ECG, it is essential to analyze the electric potential over a whole-body model [12,13].

Substantial computational cost is required to analyze the cardiac phenomena in a whole-body model, which commonly consists of a relatively large number of elements (e.g., voxel models applying finite difference methods and tetrahedral models using finite element methods). One of the limitations to processing whole-body models is the computationally expensive forward problems that need to be resolved to solve the main inverse problem [14].

To solve a single forward problem, the volume conductor model must be solved through an iterative process [15,16]. Therefore, a fast-forward problem solver can significantly contribute to accelerating the entire process. Owing to its notable success in solving finite-difference problems, a geometric multi-grid method [16] was used in our previous study [17].

A common cardiac source localization technique is based on the lead field matrix (LFM) algorithm [18,19,20,21]. An LFM is a projection matrix that defines the ratio between the cardiac current density (electric dipoles) and measured potential at the electrodes located on the body surface. The cardiac current density is then estimated through a linear inverse filter or a reconstruction algorithm based on the LFM. However, in principle, the computations required to construct an LFM associated with the test dipoles are extremely expensive.

The localization algorithm is also an important factor in the problem. The most common solution to this ill-posed problem is the minimum norm estimation [22,23]. However, this method contains non-physiological characteristics and estimates overly smeared sources [24,25,26]. Therefore, numerous studies have considered the inclusion of regularization terms [19,20,27,28]. These methods require a computational memory proportional to the square number of voxels (O(N2)). Conversely, sparse signal processing solutions have been applied to avoid estimating an overly smeared source in a bioelectromagnetic inverse problem [25,29,30]. Our focus here is the sparse-based ECG localization algorithm applied toward more accurate source estimation. In [31], numerous sparse reconstruction methods were proposed. An orthogonal matching pursuit (OMP), which is a simple, sparse reconstruction algorithm, requires a computational time and memory space proportional to the number of modeled voxels (i.e., O(N) only).

In addition to current source localization, a more detailed diagnostic based on the current direction equivalent to a vectorcardiogram is proposed [1,32]. However, a detailed analysis of this has not been demonstrated in the aforementioned studies. Hence, we propose to estimate the current location and direction simultaneously, which could be useful to improve the estimation accuracy. This information could then be useful for time-series analysis to compensate for the localization error (LE).

In the present study, we propose an ECG source localization method that combines the scalar-potential finite-difference (SPFD) method as a forward problem analysis and an OMP for solving the source localization (inverse problem). The localization performance of the proposed method is demonstrated in a whole-body human model with a spatial resolution of 2.0 × 2.0 × 2.0 mm. Moreover, we evaluate the localization performance variation corresponding to a homogeneous model, which is a model obtained by converting the anatomical human body model into a uniform homogenous tissue. We then compare the effect of the model inhomogeneity, cardiac volume, and cardiac orientation on the localization accuracy with that of the previously mentioned human body model. The novel feature of our formulation is that it estimates not only the current source location but also the current direction. For a time-domain ECG waveform, the method is augmented with a Kalman filter to improve the accuracy of the localization.

## 2. Materials and Methods

The proposed method consists of three main steps. First, we solve the ECG forward problem using an equivalent current source with an electric dipole. An LFM is then constructed through an evaluation of the forward problem, and the ECG source is estimated based on the LFM with an OMP. Finally, the estimated source location is corrected by a Kalman filter considering the estimated ECG source as multiple time data. A depiction of the proposed method is presented in Figure 1.

### 2.1. Whole-body Models

A human anatomical model named TARO, which has approximately the mean size and weight of a Japanese adult male, is used in this study [33]. This model consists of 51 anatomical tissues or organs, including the skin, muscle, bone, and heart, with a resolution of 2 mm. In TARO, the heart is represented by only one tissue; the atrium and ventricle are not explicitly identified. The electrical conductivity of the TARO model (Table 1) is computed based on the 4-Cole–Cole model at 1 Hz; the conductivity of the skin is assigned a value of 0.1 S/m [34,35]. Moreover, a homogeneous whole-body model created by changing the structure of TARO into a single homogenous tissue is also considered for comparison. The conductivity of a single homogeneous tissue is assigned as 2/3 that of the muscle tissue. Figure 2 displays the coronal and sagittal cross-sectional slices of the volume conductor models used in this study.

It is essential to assign an appropriate conductivity distribution to the volume conductor model [36]. Although extremely low skin conductivities have been reported in specific studies, in the present study, the conductivity of the skin is assigned as 0.1 S/m [35]. Lower values could correspond to the stratum corneum [37] and are thus inappropriate to be used here.

### 2.2. Solving the Forward Problem

A single electric dipole with a 2 mm (1 voxel) length, which is equivalent to the cardiac current source, is simulated. This setup is a typical assumption of an ECG forward problem [1,17]. Because the dominant frequency component of the cardiac action potential is the order of 1 Hz [1,38]. Thus, the displacement current can be ignored, which is much less than the applicable lower frequency of 100 kHz [39]. This is an estimate of the quasi-static regime in biological tissue. In this study, we compute the potentials in the analysis area using the SPFD method, which is a fast computational method for biological tissues in the frequency domain. Because of the discretization of the whole region into voxels, anatomical human body models can be easily used [40,41].

The scalar potential *φ* induced in the volume conductor is given by Poisson’s equation:(1)$$\left\{\begin{array}{c} \nabla \cdot (\sigma \nabla \phi )=-\nabla \cdot J\;in\;\Omega \\ (\sigma \nabla \phi )\cdot n_{B}=0\;on\;B_{S} \end{array}\right.,$$
where *σ*, ***J***, ***n****_B_*, *Ω*, and *B_s_* are the electrical conductivity of biological tissue, current density, unit vector outwardly normal to the body surface, volume of biological tissues, and model surface, respectively. Based on quasi-static approximation [42], the following equation can be obtained by discretizing Equation (1):(2)$$\left(\sum_{n=1}^{6}S_{n}\right)\phi_{0}-\sum_{n=1}^{6}S_{n}\phi_{n}=-\frac{\partial }{\partial t}q,$$
where *n*, *φ*_n_, *ω*, and *q* denote the node indexes, scalar potential at the *n*th node, angular frequency, charge at node “0”, respectively. *S*_n_ is edge conductance from the *n*th node to the 0 node, which is derived from the tissue conductivity of the surrounding voxels. The current flowing from one node to its neighboring node along the side of the voxels is derived by defining the scalar potentials at each node of a voxel. This branch current includes a scalar potential resulting from the applied electric charge and impedance between the nodes. Simultaneous equations are derived using Kirchhoff’s current law. The scalar potentials are solved iteratively using the successive over-relaxation (SOR) with multi-grid methods as a preconditioner [43]. There are six multigrid levels to reduce the time required for the iterative calculations, which are continued until the relative residual is less than 10^−6^.

### 2.3. Location of Cardiac Source

An LFM is a projection matrix defined by the ratio of the potential measured at the *M* electrodes and the equivalent current density at *N* known myocardium points (voxels) [19]; it is expressed through the following equation:(3)L⋅j=Φ
where **L**, ***j***, and Φ are the lead field matrix of size *M* × *3N*, current density vector of *3N* × *1*, and potential vector of *M* × *1*, respectively. We selected nine electrode (observation) points corresponding to the electrode positions of the standard 12-lead ECG located on the chest and limbs of the human model. Here, ***L*** is constructed by solving the forward problem described in Section 2.2. First, the potential vector $\Phi ={\left[\phi_{1}\;\;\phi_{2}\;\cdots \;\;\phi_{M}\right]}^{T}$ and current density in the myocardium for a single electric dipole of unit magnitude oriented in the *x*-direction as an input source are calculated using the SPFD method. The potential vector is then assigned as an LFM column vector; for the other (*N − 1*) voxels and orthogonal directions, we follow the same procedure. The LFM follows the following equation:(4)$$\begin{array}{cc} L\; & =\;[L_{1x}\cdots L_{Nx}L_{1y}\cdots L_{Ny}L_{1z}\cdots L_{Nz}]\\ & \;=\;\left[\begin{array}{ccc} \frac{\Phi^{1}}{\left\|j^{1}{}_{x_{\mathrm{anode}}}\right\|} & \cdots & \frac{\Phi^{N}}{\left\|j^{N}{}_{x_{\mathrm{anode}}}\right\|}\begin{array}{ccc} \frac{\Phi^{1}}{\left\|j^{1}{}_{y_{\mathrm{anode}}}\right\|} & \cdots & \frac{\Phi^{N}}{\left\|j^{N}{}_{y_{\mathrm{anode}}}\right\|}\begin{array}{ccc} \frac{\Phi^{1}}{\left\|j^{1}{}_{z_{\mathrm{anode}}}\right\|} & \cdots & \frac{\Phi^{1}}{\left\|j^{1}{}_{z_{\mathrm{anode}}}\right\|} \end{array} \end{array} \end{array}\right] \end{array}$$
where $j_{x}{}_{{}_{\mathrm{anode}}}$, $j_{y}{}_{{}_{\mathrm{anode}}}$, and $j_{z}{}_{{}_{\mathrm{anode}}}$ are the current density at the anode point in each base direction.

During this procedure, a set of 909 points are selected at a 6 mm interval distance, assuming that all six contact surfaces of the cardiac tissue voxel were adjacent to the cardiac tissue. Figure 3 displays the analysis points for the forward problem.

It is well known that there is no unique solution to Equation (3) because the number of electrodes is extremely small compared to the size of the current density vector. Thus, we represent the electrical activity of the heart within sparse time/spatial domains [44,45,46]. The equivalent cardiac source is localized by the OMP, which is a sparse modeling technique [31]. Table 2 presents the pseudocode of the proposed algorithm. The basic idea of this algorithm is to construct a sparse vector by selecting the most plausible basis vector from the dictionary matrix. Using this approach, a sparse current density $\hat{j}$ can be calculated from **L**, which corresponds to the dictionary matrix. The cost function can be expressed through the following equation:(5)$$\hat{j}(\hat{i})=\underset{j}{\arg\min}{\left\|L\cdot j-\Phi \right\|}_{2}^{2}\;\mathrm{subject}\;\mathrm{to}\;\mathrm{supp}\left\{\hat{j}\right\}\;\subset \;S,$$
where ***S*** is a set of support vectors.

As indicated in Table 2, we solve Equation (5) through the pseudoinverse ${\hat{L}}^{+}$ using the Moore–Penrose pseudoinverse [47].

### 2.4. Simulation Protocol

A small electric dipole 2 mm in length, when assuming the electrical activity of the heart, is placed in the cardiac tissue at a randomly selected position ***r****_t_*. The potential vector Φ and current density in the myocardium are calculated using the SPFD method. Gaussian noise is added to Φ corresponding to signal-to-noise ratios (SNRs) of 0, 10, 20, 30, and ∞ [dB]. Finally, the estimated source location $\hat{r}$ is derived from the OMP algorithm mentioned in Section 2.3. We then define LE and direction error (DE) of the current density as follows:(6)$$\mathrm{LE}=\left\|\hat{r}-r_{t}\right\|,$$
(7)$$\mathrm{DE}=\arccos\left(\frac{\hat{j}\cdot j_{t}{}_{{}_{\mathrm{anode}}}}{\left\|\hat{j}\right\|\left\|j_{t_{\mathrm{anode}}}\right\|}\right),$$
where $j_{t}{}_{{}_{\mathrm{anode}}}$ is the current density at the anode point of a test dipole. The average estimation error and standard deviation are then calculated as an evaluation metric for the localization accuracy.

### 2.5. Kalman Filtering

For a realistic cardiac source estimation, the input is ECG signals. Adapting the Kalman filter corrects the estimated localization for multiple time data such as an ECG signal using a state-space model.

Assuming that the ventricular conduction velocity is constant, the motion of the source is described by the discrete state-space model as follows:(8)$$\left\{\begin{array}{c} x(t)=x(t-1)+\mathrm{dt}\cdot u(t)+W\\ \hat{r}(t)=x(t)+V \end{array}\right.,$$
where ***x***, $\hat{r}$, ***u***, **W**, and **V** are the true source location, source location estimated by OMP, direction vector of ventricular conduction velocity, process noise, and observation noise, respectively. We assume that process and observation noise follow a Gaussian probability distribution with zero mean, and their variances are **Q** and **R**. Here, **Q** is the variance of the LE at 100 test dipole sources, and **R** is the variance of the estimated localization by randomly extracting ten databases of 500 points from the LFM. The input ***u***(t) is calculated using the current density vector $\hat{j}$ estimated by OMP and the ventricular conduction velocity and is expressed through the following equation:(9)$$u(t)=v_{\mathrm{ventricular}}\cdot \hat{j}(t)/\left\|\hat{j}(t)\right\|,$$
where $v_{\mathrm{ventricular}}$ is 1.9 m/s [1,38]. The source location ${\hat{x}}^{-}$ is predicted from the noise-free case of the described state-space model. As indicated in Table 3, an iterative procedure is applied to correct the estimated source locations using Kalman filtering.

## 3. Results

In the forward problem, the electrical potential over the whole body can be easily computed with a relatively low computational cost if the geometric multigrid method is applied. The average computational time required for each (dipole) source was approximately 60 s with an Intel Xeon Gold-6130 @ 2.10 GHz running CentOS 7.5 (32 cores). Thus, the total time required to develop the LFM was approximately 2700 min (900 locations for three rectangular directions). The post-processing time required for estimating the cardiac source location was negligible.

### 3.1. Determining Criteria for the Number of Test Dipoles

Figure 4 displays the estimation accuracy for different numbers of randomly selected test dipoles with the inhomogeneous model. The mean estimated LE and standard deviation are calculated with the estimation results of each test dipole. It can be confirmed that the mean accuracy remained at the same level after the number of test dipoles was greater than approximately 100 (at less than 2 mm). Thus, we set the number of test dipoles to 100 as the criteria and evaluated the estimation accuracy in the following sections.

### 3.2. Evaluation of Estimation Performance

Two sets of the 100 test dipoles are estimated using an LFM, constructed of the homogeneous and the inhomogeneous models. Table 4 presents the localization performance with SNR = ∞ [dB]. Figure 5 displays a visualization of the LE and DE for each of the 100 test dipoles. As indicated in Figure 5, the LEs and DEs are typically less than 10 mm and 10°, respectively, whereas clear differences are observed for certain locations.

Figure 6a displays the localization characteristics of the homogeneous model that can be localized regardless of the target position to be estimated. As indicated in Figure 6b, it is clear that the LE is maximum at the center of the cardiac tissue adjacent to the blood and lung, where the conductivity contrast is high. It can be confirmed from Figure 6a,b that the error in the homogenized model is small. Figure 7 displays the average LE for SNR = 0, 10, 20, 30, and ∞ dB. The added noise was white Gaussian noise, and 1000 patterns were repeated. As indicated in Figure 7, TARO and its homogenized model have a comparable LE for an SNR less than 10 dB.

### 3.3. Sensitivity Due to Cardiac Modeling

We next investigated the personalization of the proposed method. A demonstration was conducted while assuming cardiac tissue in different potential positions in a single beat frame. The cardiac source was estimated using the LFM of the original TARO model obtained in the previous section. This provides a better understanding of the accuracy of the proposed method at different cardiac positions and can simulate realistic situations. The cardiac tissue was rotated from −10° to 10° with respect to the z-axis. Moreover, the cardiac tissue was scaled to represent expanded/contracted volumes with a factor ranging from 90% to 110% [48].

Figure 8 and Figure 9 display the variations in the cardiac model for TARO with rotation and scaling operations. Any potential vacant volume produced after the cardiac affine transformation was filled with lung-like tissue. Figure 10 and Figure 11 display the variability of the mean estimated accuracy for 100 test dipoles with the rotated and scaled cardiac tissue. From Figure 10, it can be observed that the estimation accuracy deteriorates depending on the rotation angle. Moreover, Figure 11 indicates that there is minimal DE variability with respect to the scaling of the cardiac volume.

### 3.4. Demonstration of Source Localization Using Pseudo-ECG

Figure 12a displays the multiple cardiac sources estimated using the proposed method with a pseudo-ECG. The pseudo-ECG is a 12-lead waveform developed in our previous study [17] and different from single isolated points of dipoles used in the earlier Sections. A demonstration was conducted at 34 ms of an R-wave. The green vectors in Figure 12b illustrate the estimated direction of the electric dipole source by the proposed method. The red vectors in Figure 12b indicate how the estimated direction of the electric dipole source by the proposed method can be compensated by Kalman filtering. A more consistent track path, which is closer to the ideal pathway, can be clearly observed. Table 5 displays the localization performance of the estimation results for the OMP with and without Kalman filtering. From these results, it is clear that the Kalman filter adaptation significantly improves the estimation results.

The performance of Kalman filtering was further evaluated by adding 1000 patterns of white Gaussian noise with SNR = 20 dB to the R-wave, as indicated in Figure 13. The LE of the R-wave time step is illustrated in Figure 13. It can be observed that the LE is extremely small in the central part of the time frame; the standard deviation demonstrates relatively smaller values of the whole signal. These results indicate that Kalman filtering compensates for variations in the OMP results and provides robust and stable estimation results. The performance of Kalman filtering with the rotated and scaled cardiac models is displayed in Figure 14. These results demonstrated that the estimation accuracy depends on the cardiac state during a beating motion.

## 4. Discussion

In this study, we proposed a new method of ECG source localization that considers a realistically shaped and inhomogeneous human whole-body model based on an OMP. One feature of our method is that it estimates the source location and direction simultaneously. The estimated result was processed in the time domain using a Kalman filter to improve the localization performance.

As indicated in Figure 5, the LE was less than 10 mm for more than 90% of the source positions in the homogeneous model. Moreover, it was confirmed that the DE was approximately 10° or less. For a homogeneous model, the propagation of the potential only follows the geometry of the volume conductor, and thus, the influence of the tissue inhomogeneity on the estimation accuracy is excluded. As indicated in Figure 6b, the LE varied for different places because the tissue surrounding the cardiac tissue differs depending on the position of the estimation target. In particular, the high conductivity of the blood could result in a change in the potential propagation and consequently influence the estimation accuracy if not modeled accurately. As indicated in Figure 7, the LE was stable and more accurate when the SNR was 20 dB or greater in the proposed method for the anatomical model. This suggests that a denoizing technique is required for an SNR less than 20 dB (see [48,49] for a reduction in the Gaussian noise). However, these LEs are for single isolated time points of each dipole. Therefore, when multiple time points are used, as in ECG signals, the Kalman filter can improve the estimation, as demonstrated in Section 3.4.

Figure 10 indicates that the proposed method is sensitive to simple rotations of the cardiac tissue, which depends on the cardiac model used for constructing the LFM. Cardiac tissue is a moving object whose exact static orientation is difficult to recognize. Thus, the anatomical models are provided with static phase representation of the cardiac tissue except for the 4D extended cardiac-torso model [50]. An imaging technique that avoids spatial artifacts such as the position and angle of the cardiac tissue is necessary for personalizing the proposed method [51,52]. We applied the proposed method to multiple source localizations during cardiac activity, as displayed in Figure 12. Mislocalization can easily occur if the time series is computed as indicated by the green vector in Figure 12b. However, the physical laws, i.e., the current continuity, must be satisfied. As indicated in Table 5, the application of a Kalman filter significantly improved the estimation results, where the state-space model uses estimated directions as input. Thus, improvement of the localization is supported by the proposed method for an estimation of the current source location and direction simultaneously. Note that the estimation of the current direction is stable, unlike the location, which is influenced by the model inhomogeneity.

We compared our localization accuracy with that of a previous study [53,54]. Table 6 summarizes the localization performance of the proposed method, and that reported in previous studies. Note that the majority of recent papers [8,20,53] have considered a greater number of electrodes (e.g., 100) to improve the localization accuracy. The minimum LE for an SNR of greater than 30 dB was approximately 5.0 and 12.6 mm in the homogeneous and inhomogeneous models, respectively. The latter value is somewhat greater than 10.1 mm, as reported by Svehlikova et al. [53].

The position of test dipoles in [53] was selected just from the voxels (points) used for developing transfer matrices, whereas it was selected randomly from all voxels in the heart volume in our study. In general, the inverse problem for unknown points has a greater localization error than for known points. Moreover, the input source in [53] is 30 ms of signal for one estimation point, which may not be suitable for time-series estimation.

The minimum LE in [54] was 4.4 mm. A straightforward comparison is infeasible because the edge length of the tetrahedral torso model in [54] is 6.7 ± 1.5 mm, whereas it is 2 mm in our study. In general, models constructed with a lower resolution provide a greater estimation accuracy, although the definition of exact location inside a single voxel or tetrahedral is arbitrary. An LE less than the tetrahedral dimension could result in zero errors (e.g., [55] for localization of an electroencephalogram source). Moreover, as described in [54], the skin, fat, and muscle are homogenized as one tissue. In our previous study [56] on EEG (electroencephalography), we demonstrated that tissue inhomogeneity may cause the localization error, which is attributable to the abrupt change of electrical conductivity). Thus the inhomogeneity in this study and that in [54] are different.

The maximum error in our estimation was 54 mm, which is comparable to 60 mm, as reported in [54]. The worst estimation was observed in the left ventricle summit region and at the endocardial apex of the right ventricle in [54], whereas it was near the surface of the central part of the heart in our study. As indicated in Figure 6b, the estimation error in the corresponding region would be less than 20 mm. In our study, this difference can be attributed to the higher inhomogeneity of the conductivity. In general, our computational results provide a comparable or somewhat better accuracy than the previous study. This computational approach is thus applicable not only for accurate localization but also for the design of wearable ECG sensing systems. The limitation of this study is that, as is similar to other computational studies, personalized human body models are needed to apply measured results. Thus, further study on the morphing human body models would be needed [57,58].

## 5. Conclusions

A localization method for determining cardiac sources was proposed by combining an electrical analysis of a realistic human body as the forward problem and a sparse reconstruction method as the inverse problem. For a 12-lead electrocardiogram system, a sensitivity analysis of the localization to the cardiac volume, tilted angle, and model inhomogeneity was conducted. Once an LFM was constructed, the estimation of the source location was virtually instantaneous.

For a noise-free condition, the average LE for an isolated time point was 12.6 mm, which is comparable to or somewhat superior to that reported in a previous study. Time-series source localization with Kalman filtering for the estimated location in terms of estimated current direction and location demonstrated that source mislocalization could be compensated, suggesting the effectiveness of the proposed method. For the ECG R-wave, the mean distance error was reduced to less than 7.3 mm using the proposed method. This highly accurate estimation was achieved because the proposed approach uses an estimation of the current direction, which is less sensitive to different error sources. Considering the physical properties of the human body with Kalman filtering enables highly accurate estimation of the cardiac electric signal source location and direction. Our proposal is applicable to the electrode configuration in wearable sensing systems where non-conventional locations would be more essential.

## Figures and Tables

**Figure 1 sensors-21-04275-f001:**
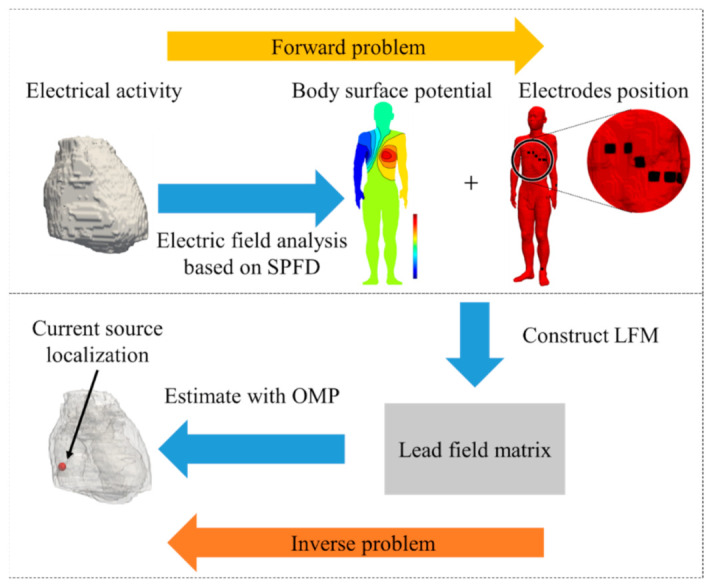
Schematic explanation of the proposed method for cardiac current source estimation.

**Figure 2 sensors-21-04275-f002:**
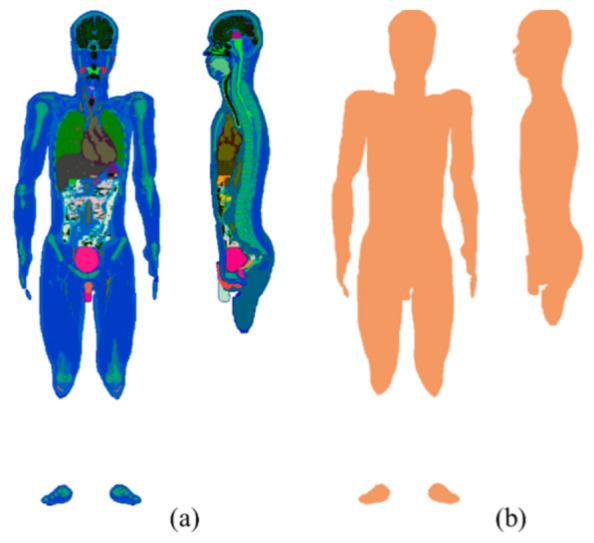
Coronal (left) and sagittal (right) cross-sectional slices of (**a**) TARO and (**b**) its homogenized models.

**Figure 3 sensors-21-04275-f003:**
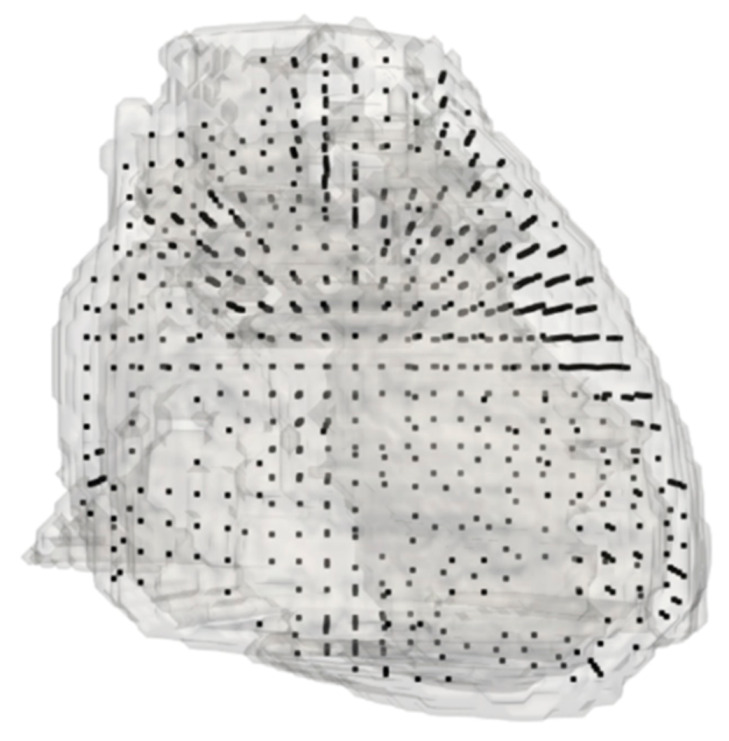
All points of forward problem analysis. The black dots represent the positions of the analysis points over a volume rendering of the TARO heart model.

**Figure 4 sensors-21-04275-f004:**
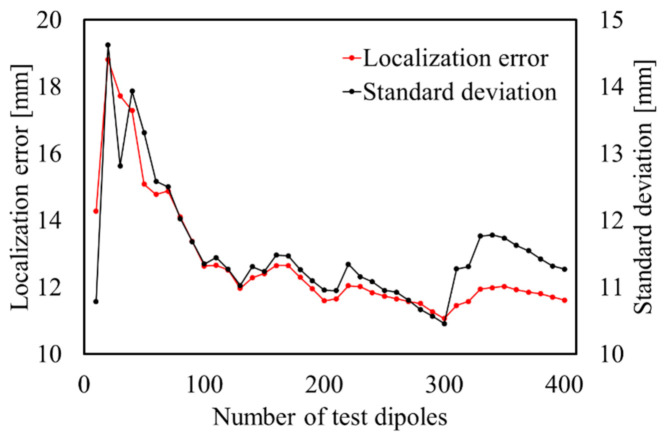
Estimated accuracy for each number of test dipoles with SNR = ∞ dB.

**Figure 5 sensors-21-04275-f005:**
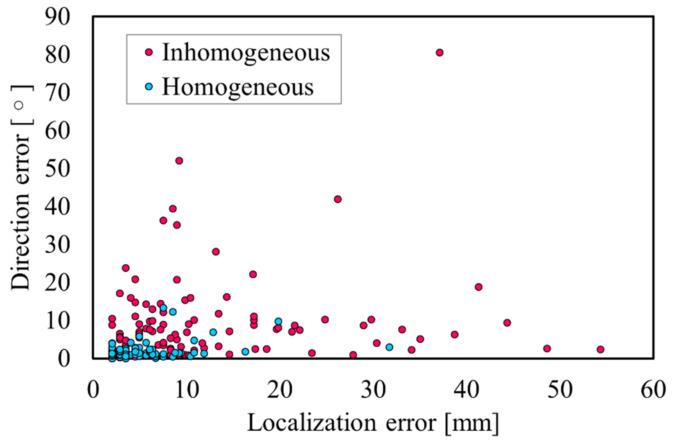
Relationship between LE and DE for each model with SNR = ∞.

**Figure 6 sensors-21-04275-f006:**
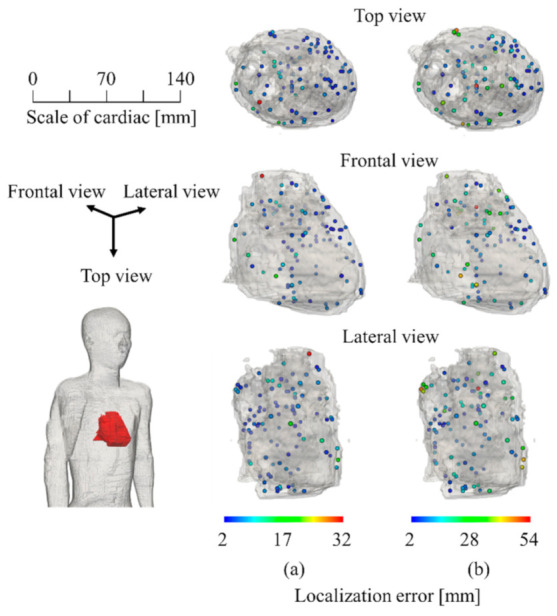
Visualized LE for each of 100 test dipoles: (**a**) homogeneous and (**b**) inhomogeneous.

**Figure 7 sensors-21-04275-f007:**
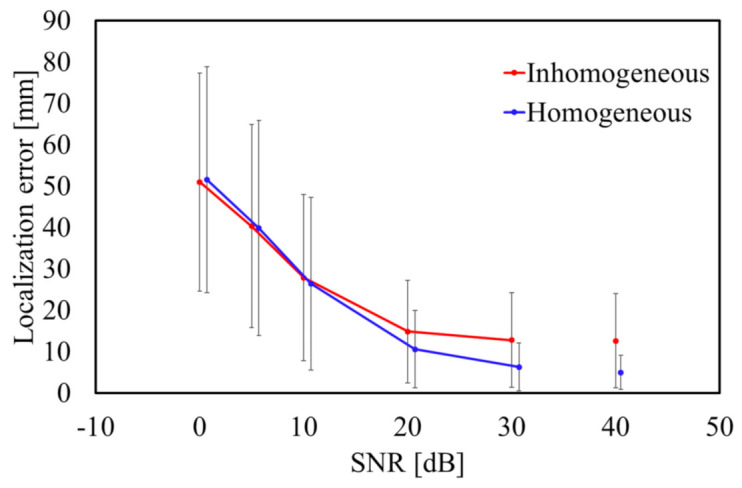
Average LE for different SNRs. Error bars represent the standard deviation.

**Figure 8 sensors-21-04275-f008:**
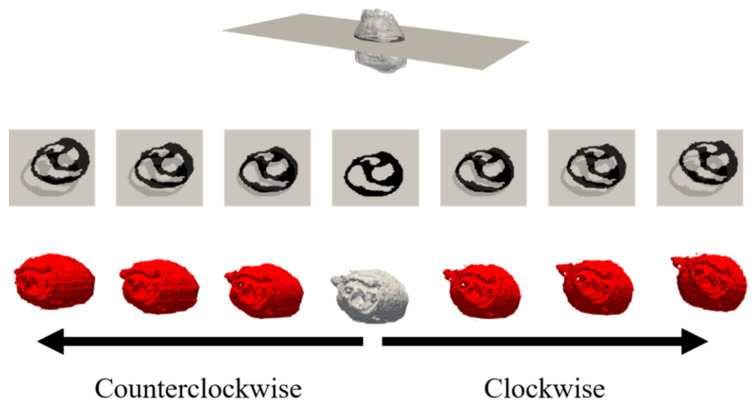
Variations of TARO cardiac model: rotation with ±2, ±5, and ±10 degrees displayed in ascending order from left to right. Upper images are axial cross-sectional slices, and bottom images are volume renderings.

**Figure 9 sensors-21-04275-f009:**
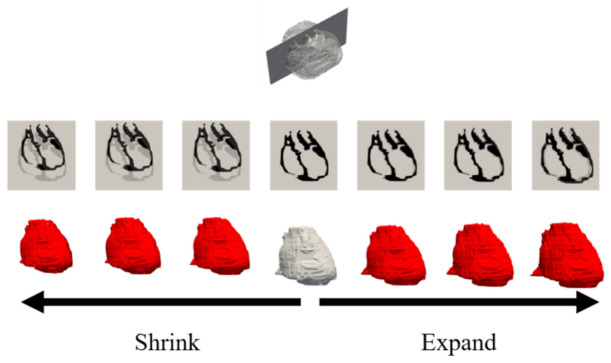
Variations in TARO cardiac model: scaling with a value of ±2%, ±5%, and ±10% of the original volume displayed in ascending order from left to right. Upper images are coronal cross-sectional slices, and bottom images are volume renderings.

**Figure 10 sensors-21-04275-f010:**
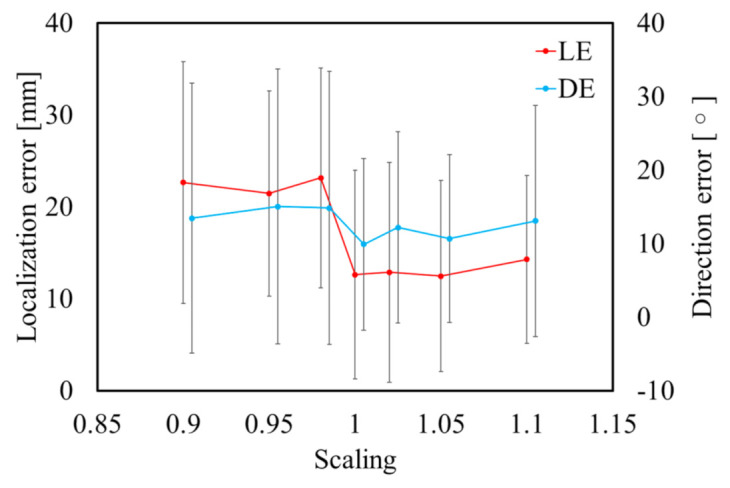
Variability of estimation accuracy for each rotation angle of heart with SNR = ∞. Error bars represent standard deviation over 100 test dipoles.

**Figure 11 sensors-21-04275-f011:**
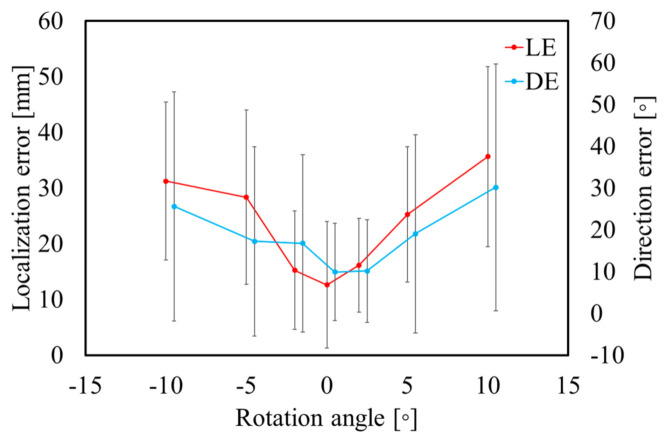
Variability of estimation accuracy for each scaling of heart with SNR = ∞. Error bars represent standard deviation over 100 test dipoles.

**Figure 12 sensors-21-04275-f012:**
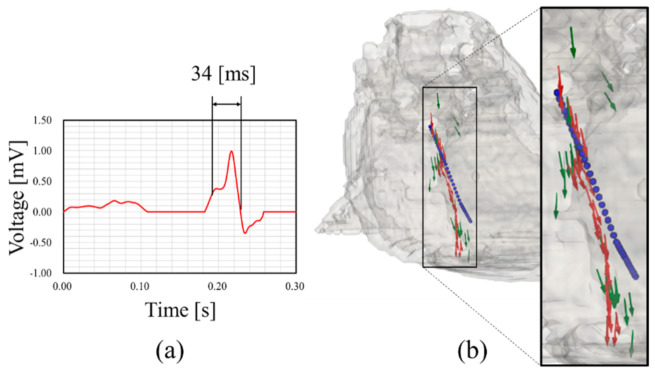
(**a**) ECG waveform in lead II constructed based on [16] and (**b**) multiple source localization corresponding to R-wave. Blue dots represent an ideal pathway. Red and green color vectors represent the estimated direction of electric dipole source at an elapsed time corresponding to OMP with and without Kalman filtering, respectively.

**Figure 13 sensors-21-04275-f013:**
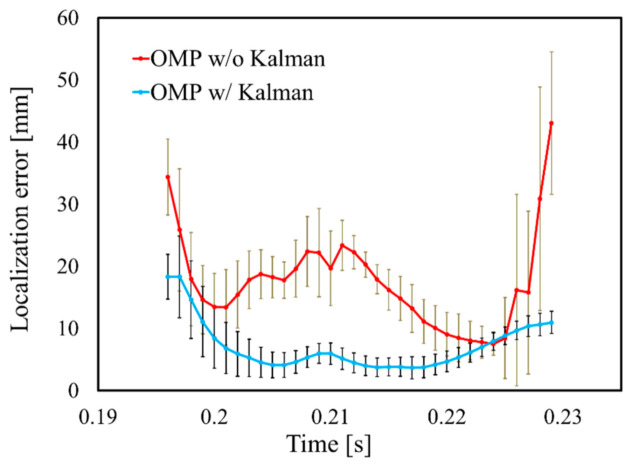
LE on R-wave time step with SNR = 20 dB. Error bars represent standard deviation.

**Figure 14 sensors-21-04275-f014:**
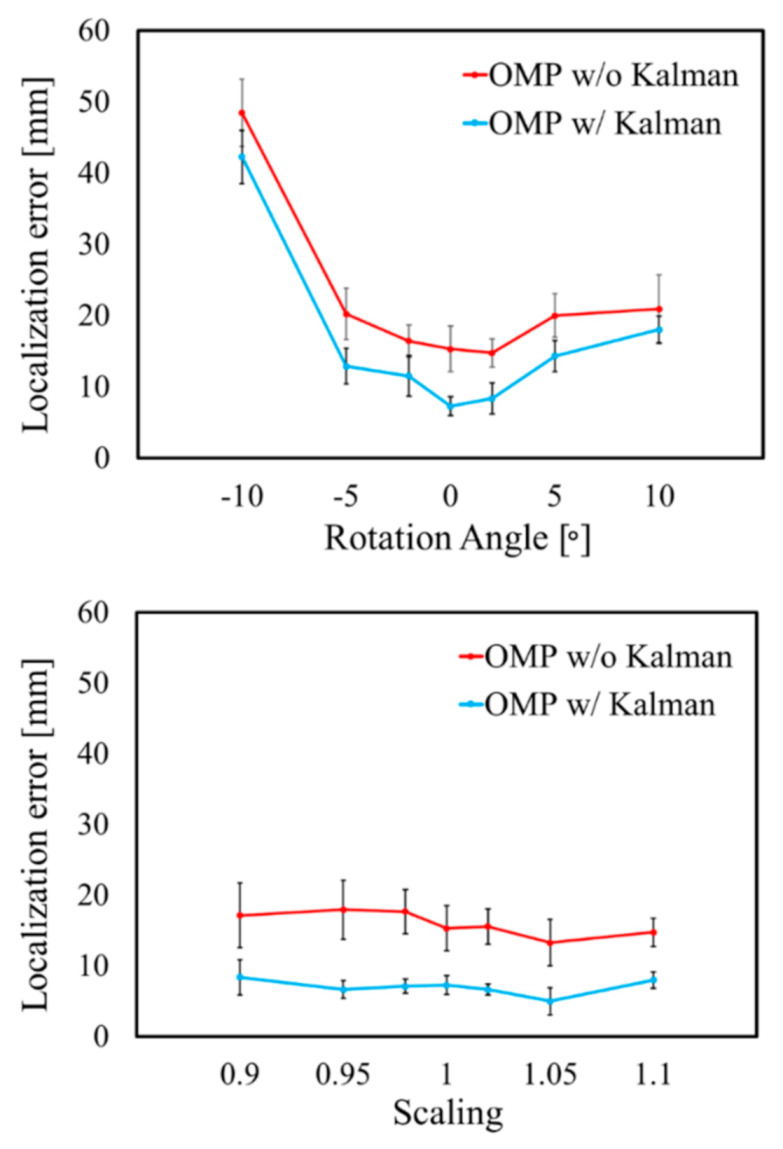
LE on R-wave for each rotation angle and scaling of heart. Error bars represent standard deviation.

**Table 1 sensors-21-04275-t001:** Conductivity of human tissues.

Tissues	Conductivity (S/m)	Tissues	Conductivity (S/m)
Adrenal	0.20	Hypothalamus	0.02
Air	0.00	Internal air	0.00
Bile	1.40	Kidneys	0.05
Bladder	0.20	Lens	0.30
Blood	0.70	Ligaments	0.25
Bone (cancellous)	0.07	Liver	0.02
Bone (cortical)	0.02	Lung	0.20
Cartilage	0.15	Muscle	0.20
Cerebellum	0.04	Nerve	0.01
Cerebrospinal fluid	2.00	Pancreas	0.50
Colon	0.01	Seminal capsule	0.20
Content of the large intestine	0.20	Skin	0.10
Content of the small intestine	0.20	Small intestine	0.50
Content of the stomach	0.20	Spleen	0.03
Cornea	0.40	Stomach	0.50
Corpus spongiosum	0.20	Tendon	0.25
Diaphragma sellae	0.20	Testicle	0.20
Duodenum	0.50	Testis prostate	0.40
Esophagus	0.50	Thalamus	0.02
Fat	0.04	Thyroid thymus	0.50
Gall bladder	0.90	Tongue	0.25
Glandula pinealis	0.02	Tooth	0.02
Glandula salivaria	0.20	Trachea	0.30
Glandula pituitaria	0.02	Urine	0.70
Gray matter	0.02	Vitreous humor	1.50
Heart	0.05	White matter	0.02

**Table 2 sensors-21-04275-t002:** Pseudocode of proposed algorithm using matching pursuit.

Description	Code
Set L as in Equation (4)	for *i* = 1:3*N*
	$corrcoef(i)=\frac{\Phi \times L_{i}}{\left\|\Phi \right\|\left\|L_{i}\right\|}$
	end for
Estimated source location:	$\hat{i}=\underset{i}{\arg\;\max}\;corrcoef(i)$
Support vector:	$\hat{L}=[L_{\hat{i}x},\;L_{\hat{i}y},\;L_{\hat{i}z}]$
Current density:	$\hat{j}(\hat{i})=\;{\hat{L}}^{+}\cdot \Phi$

**Table 3 sensors-21-04275-t003:** Pseudocode for estimated localization correction using Kalman filtering.

Description	Code
Initialize variance-covariance matrix:	P(0)=Q
Initialize quantity of state:	${\hat{x}}^{-}(0)=[\mathrm{AV}-\mathrm{node}\;\mathrm{location}]$
	for t = 1:N
Prediction step:	${\hat{x}}^{-}(t)={\hat{x}}^{-}(t-1)+\mathrm{dt}\cdot u(t)$
	$P^{-}(t)=P(t-1)+dt^{2}\cdot Q$
Update step:	$K(t)=P^{-}(t)/P^{-}(t)+R$
	$P(t)=(I-K(t))P^{-}(t)$
	$\hat{x}(t)={\hat{x}}^{-}(t)+K(t)(\hat{r}(t)-{\hat{x}}^{-}(t))$
	end for

**Table 4 sensors-21-04275-t004:** Statistics of localization performance with SNR = ∞.

	LE (mm)	DE (◦)
Homogeneous	5.01 ± 4.07	1.91 ± 2.09
Inhomogeneous	12.64 ± 11.35	9.93 ± 11.67

**Table 5 sensors-21-04275-t005:** Localization performance error with and without Kalman filtering.

	RMSE			Mean Distance Error (mm)
	x-Axis	y-Axis	z-Axis	
OMP w/o Kalman	4.63	10.37	13.19	15.30
OMP w/ Kalman	3.31	6.12	4.40	7.26

**Table 6 sensors-21-04275-t006:** Localization performance and specifications of the proposed method and related work.

	(Proposed)	Svehlikova, 2018. [53]	Potyagaylo, 2016. [54]
Source	dipole	dipole	double layer
Electrode	9	100	9
Resolution (mm)	2.0	-	6.7 ± 1.5
SNR (dB)	∞	∞	∞
Localization error (mm)	12.64 ± 11.35	10.13 ± 5.13	4.4 ± 5.4
Number of tissue	51	12	12

## Data Availability

Anatomical human body model is available from National Institute of Information and Communications Technology, Japan (https://emc.nict.go.jp/bio/data/index_e.html, accessed on 21 June 2021). Detailed information of current source location on cardiac tissue using this computation is available from the corresponding author (AH) on request.

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
