# Peer review of "ECG Localization Method Based on Volume Conductor Model and Kalman Filtering"

_sensors, 2021, doi:10.3390/s21134275_

Round 1

Reviewer 1 Report

Overall, this is a well-written article with sufficient analysis. The article is about compensating mislocation of ECG using Volumn Conductor Model and Kalman Filtering.

In Forward Probelm, the author use SPFD and in Inverse Problem, LFM to reconstruct the ECG Location. It is known that inverse Problem is ill-posed problem. So the quality of the ECG location will depend on how to solve Inverse problem well. The main point of this article is about reducing the mislocalization of ECG by the proposed model.  Especially, Kalman filter is a key point to reduce estimation error in this study.

The authors well explained about the difference in localization performance error with and without the Kalman filter.

[Comparision with previous work]

To strengthen this study, please consider some graphical things like chart and graph to prove that the proposed method is better than previous work. Result with better performance than the previous method is written only in line 1, page 15.

It would be more clear to compare the proposed method with some previous works under limitations.

[Presentational problems]

Some revisions needed, like font size of legend and label in graph

p.13, figure13: The font size of axis label is too big compared to the font size of other graph.

The format of pseudo code in Table 2 and Table 3 needs to be arranged.  Also, I would recommend to separate the code and the description. It would be helpful to the readers.

On page 3, sentences are too long in lines 2-6. It would be more comprehensible if you break them down into 2-3 shorter sentences. 

Author Response

Thank you for taking your time to review our manuscript. Based on your suggestion, we have improved our manuscript. Please find the attached file.

Reviewer 2 Report

I find the paper very well presented and proposes a very interesting (and difficult) problem. However, I have some doubts and concerns:

  • pp 9, figure 4. What is the reason for the evolution of the plot? At first thought, one might think that the curve should be smoother with a growing number of test dipoles, but it is not. Is it related to the parameter (conditions) of the problem?
  • pp 10, section 3.3. I understand the goal of the analysis is to characterize/evaluate a dynamic body, that is, one that gets lungs inflated and heart pumping flow. How representative are rotations/scaling are? I mean, in a real body I understand that deformation of the different tissues is not similar, so the actual rotation, translation and scaling are not uniform
  • pp 13, section 3.4. The demonstration of the method is performed using a 34 ms R-wave. While it is fine for understanding the local functioning of the algorithms, could you provide a longer and realistic one? Is this method is going to be usable in a real application, the localization should perform well during the whole ECG analysis. Otherwise, how do you propose to use the method?
  • I understand that this is a simulated scenario. But in order to evaluate the goodness of the solution (that is, it is not yet a different number crunching algorithm), a more in depth discussion of how close to be possible to use with a real body is would be needed. For example, a 2mm resolution model of the subject would be need, wouldn't it? Is there a feasible method to create the model? Etc.

Finally, some minor suggestions:

  • pp 5, figure 2b. I might represent it a little bit darker. When printed on paper it is not very viewable
  • pp 14, section 4. There is much information, which makes it hard to follow. I would add a table summarizing the comparison between the different localization methods (error, features, etc)
  •  

Author Response

Thank you for taking your time to review our manuscript. We have revised the manuscript based on your suggestion. Please find the attached file for our response.
